# Evaluation of the Performances of Radar and Lidar Altimetry Missions for Water Level Retrievals in Mountainous Environment: The Case of the Swiss Lakes

**Frédéric Frappart** [1,*] , **Fabien Blarel** [1] , **Ibrahim Fayad** [2] , **Muriel Bergé-Nguyen** [1,3] , **Jean-François Crétaux** [1,3] , **Song Shu** [4] , **Joël Schregenberger** [5,6] **and Nicolas Baghdadi** [2]

1    LEGOS, Université de Toulouse (CNES/CNRS/IRD/UPS), 14 Avenue Edouard Belin, 31400 Toulouse, France; fabien.blarel@legos.obs-mip.fr (F.B.); muriel.berge-nguyen@cnes.fr (M.B.-N.); jean-francois.cretaux@legos.obs-mip.fr (J.-F.C.)
2    CIRAD, CNRS, INRAE, TETIS, University of Montpellier, AgroParisTech, CEDEX 5, 34093 Montpellier, France; ibrahim.fayad@inrae.fr (I.F.); nicolas.baghdadi@teledetection.fr (N.B.)
3    CNES, 18 Avenue Edouard Belin, CEDEX, 31401 Toulouse, France
4    Department of Geography and Planning, Appalachian State University, Boone, NC 28608, USA; shus@appstate.edu
5    Eidg. Departement für Umwelt, Verkehr, Energie und Kommunikation UVEK, Bundesamt für Umwelt BAFU, Abteilung Hydrologie, 3003 Bern, Switzerland; joel.schregenberger@students.unibe.ch
6    Faculty of Humanities, Institut für Sprachwissenschaft, University of Bern, Länggassstrasse 49, 3012 Bern, Switzerland
*    Correspondence: frederic.frappart@legos.obs-mip.fr

**Abstract:** Radar altimetry is now commonly used to provide long-term monitoring of inland water levels in complement to or for replacing disappearing in situ networks of gauge stations. Recent improvements in tracking and acquisition modes improved the quality the water retrievals. The newly implemented Open Loop mode is likely to increase the number of monitored water bodies owing to the use of an a priori elevation, especially in hilly and mountainous areas. The novelty of this study is to provide a comprehensive evaluation of the performances of the past and current radar altimetry missions according to their acquisition (Low Resolution Mode or Synthetic Aperture Radar) and tracking (close or open loop) modes, and acquisition frequency (Ku or Ka) in a mountainous area where tracking losses of the signal are likely to occur, as well as of the recently launched ICESat-2 and GEDI lidar missions. To do so, we evaluate the quality of water level retrievals from most radar altimetry missions launched after 1995 over eight lakes in Switzerland, using the recently developed ALtimetry Time Series software, to compare the performances of the new tracking and acquisition modes and also the impact of the frequency used. The combination of the Open Loop tracking mode with the Synthetic Aperture Radar acquisition mode on SENTINEL-3A and B missions outperforms the classical Low Resolution Mode of the other missions with a lake observability greater than 95%, an almost constant bias of ($-0.17 \pm 0.04$) m, a RMSE generally lower than 0.07 m and a R most of the times higher than 0.85 when compared to in situ gauge records. To increase the number of lakes that can be monitored and the temporal sampling of the water level retrievals, data acquired by lidar altimetry missions were also considered. Very accurate results were also obtained with ICESat-2 data with RMSE lower than 0.06 and R higher than 0.95 when compared to in situ water levels. An almost constant bias ($0.42 \pm 0.03$) m was also observed. More contrasted results were obtained using GEDI. As these data were available on a shorter time period, more analyses are necessary to determine their potential for retrieving water levels.

**Keywords:** radar and laser altimetry; lakes; validation

## 1. Introduction

    Lakes and reservoirs are considered as sentinels, integrators, and regulators of climate change owing to rapid responses of their physical, chemical, and biological properties to

climate-induced variations [1–3]. Lake properties are affected by climate related changes (e.g., variations in solar radiation, air temperature, and rainfall) at multiple time-scales ranging from a short and intense storm event to seasonal changes to longer-term interdecadal variations [4]. Various anthropogenic stresses also strongly modified lakes ecosystems [5,6]. Variations in lake water levels are directly reflecting the impact of climate change and anthropogenic actions such as of strong rain event, a long-lasting drought, or excessive pumping in the lake or its underneath groundwater for irrigation or human consumption purposes.

Decline in in situ water stage gauge numbers has been reported worldwide, including for lakes [7,8]. In this context, radar altimetry has demonstrated strong capabilities in the monitoring of inland water bodies, and especially of lakes [9,10]. Advances in sensor properties, such as the acquisitions at Ka-band on SARAL in 2013, which provided measurements with a smaller footprint and a higher bandwidth [11]; the generalization of the Synthetic Aperture Radar (SAR) mode [12] on the recent radar altimetry missions (i.e., Cryosat-2, Sentinel-3, and Sentinel-6/Jason-Continuity Service (CS)) to reduce the size of the altimeter footprint in the along-track direction; the Open-Loop (OL) or Digital Elevation Model (DEM) tracking mode [13,14] developed to limit the loss of data over hilly areas; and development of refined processing techniques (i.e., the use of the high-rate altimetry data, and of more robust waveform retracking algorithm than the ocean ones based on the modeling of the response of an ocean surface or Brown model [15], both increased the number of observations of the inland water bodies each cycle) allowed the monitoring of small size lakes of width along the track lower than 500 m [16,17]. As high precision radar altimeters have been operating for more than 20 years, long-term time-series, combining data from several radar altimetry missions operating on the same (i.e., 10-day repeat orbit from Topex-Poseidon, Jason-1, 2, 3, and Sentinel-6/Jason-CS, 35-day repeat orbit from ERS-1, 2, ENVISAT, and SARAL, 27-day repeat orbits from Sentinel-3A and B) or on several orbits, were produced to monitor the changes in lake levels and storage in response to climate change and anthropogenic effects (e.g., [18–21]).

To have confidence in radar altimetry-based water levels, calibration/validation (cal/val) needs to be performed to assess each mission bias, inter-mission biases (when combining data from different radar altimeters), and to determine the quality of the datasets through measures such as root-mean-square error (RMSE) and correlation coefficient (R). Even though numerous cal/val permanent sites are operating over the ocean, less cal/val facilities are available for inland water. Lake Issykkul, in Kyrgyzstan (Central Asia), is, to our knowledge, a unique site where comparisons between in situ data, Global Navigation Satellite Systems (GNSS), and radar altimetry-based water levels have been performed on regular basis since mid-2000 [22–24]. Recently, a study evaluated the performances of 11 radar altimeter (RA) missions, from GEOSAT to SENTINEL-3A, over large lakes with areas ranging from 487 to 81,935 km$^2$ [25].

Since radar altimetry collects the elevation measurements of Earth's surface at the nadir direction, only a small portion of the Earth surface is covered by the altimetry tracks. According to [26], current altimetry missions on repeat orbit, not considering Cryosat-2, are only able to monitor less than half of the total number of lakes with an area larger than 100 km$^2$. To increase the number of lakes where water levels can be retrieved, lidar altimetry data from ICESat missions were increasingly used because of their small footprint (~70 m for ICESat/GAS and ~17 m for ICESat-2) and long-period/drifting orbit which allows to observe a larger part of the Earth's surface at the expense of temporal frequency [27–29]. Two new satellite lidar altimetry missions were launched in 2018: ICESat-2 and the Global Ecosystem Dynamics Investigation Lidar (GEDI) whose main goals are to monitor the elevation changes of the Greenland and Antarctic ice sheets and to provide observations of forest vertical structures, respectively.

In this study, an evaluation of the performance of most of the high-resolution radar altimetry missions (i.e., Jason-1/2/3, ERS-2, ENVISAT, SARAL, and SENTINEL-3A and B) is performed over eight Swiss lakes, with an emphasis on the recently launched SENTINEL-

3A and B missions. The study area was chosen because of i) the presence of a high quality, continuous, and leveled dataset of water levels covering the whole 1995–2020 study period, and ii) the strong topography surrounding the lakes which will allow to assess the impact of the OL tracking mode on radar altimetry-based water level measurements. A complementary cal/val study was also performed on the ICESat-2 and GEDI data available in the study area to determine their advantages and drawbacks for the monitoring of lake levels.

## 2. Materials and Methods

### 2.1. Study Area

The study area is composed of 10 lakes located in Switzerland (Figure 1a,b). The altitude the lakes basin and their soundings range from 300 to above 4000 m.a.s.l in a rugged and mountainous area (Figure 1c). As most of the Swiss Alpine lakes, the lakes considered in this study are characterized by an elongate shape and an axial outflow and surrounded with steep lateral mountain slopes [30].

### 2.2. Datasets

#### 2.2.1. Radar Altimetry Data

Data from the following high precision radar altimetry missions were used in this study: ERS-2, ENVISAT, Jason-1/2/3, SARAL (operating in Low Resolution Mode—LRM), SENTINEL-3A and 3B (operating in SAR mode). These missions are acquiring data on three different nominal orbits. Their major characteristics are summarized below:

1. Ten-day repeat orbit period missions: Jason-1, Jason-2, and Jason-3

Jason-1, Jason-2, and Jason-3 were placed on a 1336 km of altitude and 66° of inclination orbit with a 10-day repeat cycle, and an equatorial ground-track spacing of about 315 km, formerly used by the Topex/Poseidon mission.

Jason-1 is a joint mission between the Centre National d'Etudes Spatiales (CNES) and the National Aeronautics and Space Agency (NASA) launched on 7 December 2001. Jason-1 payload is composed of the bi-frequency altimeter operating at C (5.3 GHz) and Ku (13.575 GHz)-bands Poseidon-2 from CNES, of the Jason Microwave Radiometer from NASA and of a triple system for precise orbit determination: DORIS instrument from CNES, Black Jack Global Positioning System receiver from NASA, and a Laser Retroflector Array (LRA) from NASA/Jet Propulsion Laboratory (JPL) [31]. Jason-1 remained in its nominal orbit until 26 January 2009 and was decommissioned on 21 June 2013.

Jason-2 is a joint mission between CNES, the European Organization for the Exploitation of Meteorological Satellites (EUMETSAT), NASA and the National Oceanic and Atmospheric Administration (NOAA) launched on 20 June 2008. Jason-2 payload consists of the Poseidon-3 radar altimeter developed by CNES, the Advanced Microwave Radiometer (AMR) from JPL/NASA, and a triple system for precise orbit determination: the real-time tracking system Détermination Immédiate d'Orbite par Doris embarqué (DIODE) of Doppler Orbitography and Radio-positioning Integrated by Satellite (DORIS) instrument from CNES, a Global Navigation Satellite System (GNSS) receiver and a LRA from NASA/JPL. Poseidon-3 radar altimeter is a two-frequency solid-state altimeter, operating at Ku and C-bands designed to accurately determine the distance between the satellite and the surface or range and to provide the ionospheric correction to the range over the ocean [32]. Jason-2 remained in its nominal orbit until 3 July 2016 and was decommissioned on 4 October 2019.

Jason-3 mission is a joint mission from CNES, EUMETSAT, NASA, and NOAA. It was launched on 17 January 2016 [33]. Its payload consists of the bi-frequency Poseidon-3B radar altimeter operating at Ku and C-bands, a Precise Orbit Determination (POD) package composed of a GPS and a DORIS receiver, and a LRA from NASA/JPL.

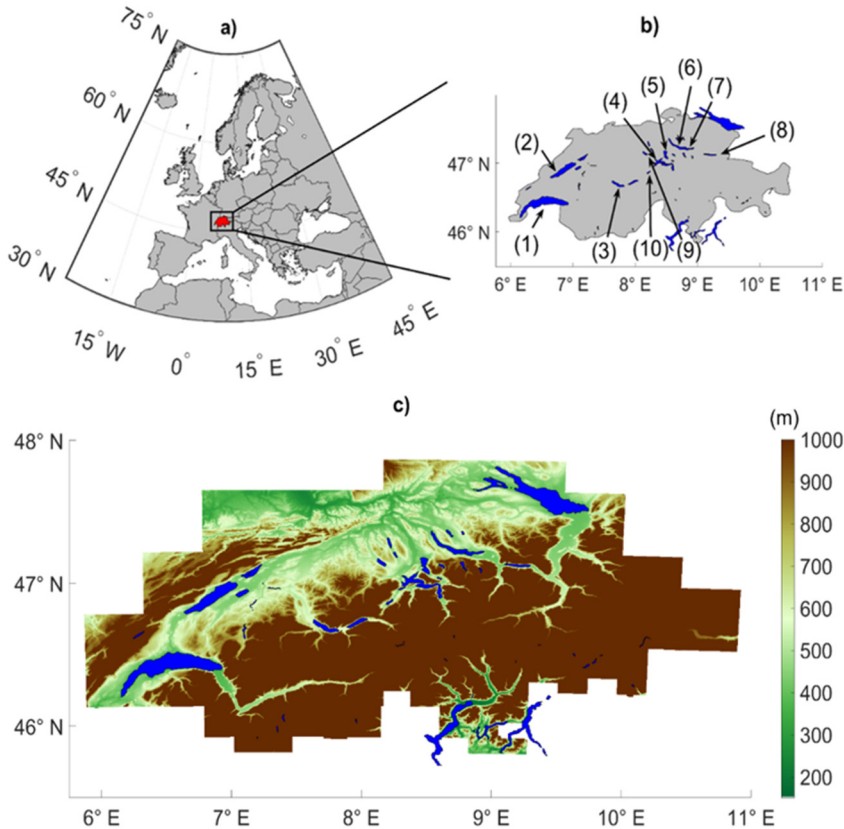

**Figure 1.** (**a**) The study area is located in Switzerland, in Western Europe, (in red) (**a**) and is composed of the following lakes (in blue) (**b**): Lake Geneva (1), Lake Neuchâtel (2), Lake Thun (3), Lake Lucerne (4), Lake Zug (5), Lake Zürich (6), Lake Zürich (Obersee) (7), Walensee (8), Lake Sempach (9), Lake Sarnen (10). (**c**) These lakes are located in the Swiss Alps mountainous region where the altitude ranges from 300 to more than 4000 m.a.s.l. The Digital Elevation Model (DEM) used as background is the Digital Height Model DHM25 at 200 m of spatial resolution made available by SwissTopo [34].

## 2. Thirty-five-day repeat orbit period missions: ERS-2, ENVISAT, and SARAL

ERS-2, ENVISAT, and SARAL were placed on a ~790 km of altitude and a 98.54° inclination sun-synchronous orbit with a 35-day repeat cycle and an equatorial ground-track spacing of about 85 km.

ERS-2 was launched on 25 April 1995 by the European Space Agency (ESA) as ERS-1 follow-on mission. The satellite carried, among other sensors, a radar altimeter (RA) operating at Ku-band (13.8 GHz) [35].

ENVISAT mission was launched on 1 March 2002 by ESA. Its payload was composed of 10 instruments including the advanced radar altimeter (RA-2). It was based on the heritage of the sensor on-board the ERS-1 and 2 satellites. RA-2 was a nadir-looking pulse-limited RA operating at two frequencies at Ku- (13.575 GHz), as ERS-1 and 2, and S-(3.2 GHz) bands [36]. ENVISAT remained on its nominal orbit until October 2010 and its mission ended on 8 April 2012. RA-2 stopped providing valid data at S-band in January 2008.

SARAL is a joint mission between CNES and the Indian Space Research Organization (ISRO), launched on 25 February 2013. Its payload is composed of the AltiKa radar altimeter operating at Ka-band (35.75 GHz), the bi-frequency radiometer, and a triple system for precise orbit determination: the real-time tracking system DIODE of the DORIS instrument, a LRA, and the Advance Research and Global Observation Satellite (ARGOS-3) [37]. It remained on its nominal orbit until July 2016.

## 3. Twenty-seven-day repeat orbit period missions: SENTINEL-3A and SENTINEL-3B

Sentinel-3 is a mission developed by ESA in the framework of the COPERNICUS program. Two satellite were already launched: SENTINEL-3A on 16 February 2016 and SENTINEL-3B on 25 April 2018. They are placed on an orbit at 814.5 km altitude and a 98.65° inclination sun-synchronous orbit with a 27-day repeat cycle and an equatorial ground-track spacing of about 105 km. SENTINEL-3A and SENTINEL-3B are on the same orbit with a phase difference of 180°. The satellites payloads is composed of SRAL (SAR Radar ALtimeter), a dual-frequency SAR altimeter (Ku-band at 13.575 GHz and C-band at 5.41 GHz), a Microwave Radiometer (MWR) instrument for wet path delay measurements and a triple system for precise orbit determination: a POD including a GPS receiver, a LRA and a DORIS instrument [38].

The main characteristics of the data used in this study are summarized in Table 1. They are made available by CTOH [39].

**Table 1.** Main characteristics of the radar altimetry data from the high-precision missions used in this study. The start and end dates are given for the nominal orbit of the different radar altimetry missions.

| Altimetry Mission | Jason-1 | Jason-2 | Jason-3 | ERS-2 | ENVISAT | SARAL | SENTINEL-3 |
|---|---|---|---|---|---|---|---|
| Start | 01/2002 | 07/2008 | 02/2019 | 05/1995 | 05/2002 | 03/2013 | 02/2016 (A) 04/2018 (B) |
| End | 01/2009 | 10/2016 | On-going | 11/2003 | 10/2010 | 07/2016 | On-going (A and B) |
| GDR | E | D | D | CTOH [40] | v2.1 | T | |
| Along-track sampling | 20 Hz | 20 Hz | 20 Hz | 20 Hz | 18 Hz | 40 Hz | 20 Hz |
| Retracker | ICE | ICE | ICE | ICE-1 | ICE-1 | ICE-1 | OCOG |
| $\Delta R_{iono}$ | global ionospheric map (GIM) [41]-based [1] | | | | | | |
| $\Delta R_{dry}$ | European Centre for Medium-Range Weather Forecasts (ECMWF)-based using Digital Elevation Model (DEM) | | | ECMWF-based using h | | ECMWF-based using DEM | |
| $\Delta R_{wet}$ $\Delta R_{solid\ Earth}$ $\Delta R_{pole}$ | ECMWF-based using DEM Based on Cartwright et al. [42] Based on Wahr et al. [43] | | | | | | |

[1] NIC09-based corrections were used before 09/1998 for ERS-2 as GIM-based ones were not available.

2.2.2. Lidar Altimetry Data

1.    ICESat-2

ICESat-2 mission was launched on 15 September 2018 by NASA. It is placed on an orbit at ~500 km altitude and a 92° inclination sun-synchronous orbit with a 91-day repeat cycle and an equatorial ground-track spacing of about 28.8 km. It carries onboard Advanced Topographic Laser Altimeter System (ATLAS), a low energy multibeam laser operating at a wavelength of 532 nm (green) in conjunction with single-photon sensitive detectors to determine the range. The along-track distance between two adjacent shots is 0.7 m owing to the 10 kHz pulse repetition rate [44]. A diffractive optical element is used to diffract each pulse into three pairs of beams, located 90 m across-track and identified by its orientation (left—L; or right—R) and its spot number (1 to 3), (i.e., GT1L/GT1R, GT2L/GT2R, and GT3L/GT3R). The multi-beam configuration allows to estimate the local terrain slope and to detect surfaces with low and high reflectivity, defining a strong and a weak beam in each pair, which varies with the ATLAS orientation. The ICESat-2 footprint has a diameter ~17 m which could increase up to ~20 m due an energy decrease after the 3-year of the nominal mission [45]. In this study, the ATLAS/ICESat-2 Level3A (ATL13) product, providing along-track heights of lakes, rivers, and wetlands with reference to EGM2008 is used [46]. It is made available by the National Snow and Ice Data Centre (NSIDC) at [47].

Over our study areas, there are between 3 and 12 acquisition series per lake using ICESat-2 data (Table 2).

**Table 2.** ICESat-2 acquisition dates where valid data were obtained over the studied lakes.

| Lake | Dates |
|---|---|
| Geneva | 3 December 2018; 29 December 2018; 27 January 2019; 2 February 2019; 25 February 2020; 30 March 2019; 28 April 2019; 3 August 2019; 25 August 2019; 27 September 2019; 24 November 2019; 23 December 2019 |
| Neuchâtel | 3 March 2019; 26 March 2019; 2 June 2019; 25 June 2019; 23 December 2019 |
| Thun | 29 November 2018; 27 February 2019; 22 March 2019; 26 September 2019; 18 October 2019 |
| Lucerne | 17 December 2018; 23 February 2019; 18 March 2019; 25 May 2019; 17 June 2019; 24 August 2019; 15 September 2019; 25 October 2019 |
| Zug | 25 May 2019; 15 March 2020; 22 May 2020 |
| Zürich | 02/05; 04/05; 11/05; 08/06; 04/07; 01/09; 23/09 |

2. GEDI

The Global Ecosystem Dynamics Investigation (GEDI) is the latest operational full waveform (FW) spaceborne LiDAR system, and has been acquiring data since April 2019 on board the International Space Station (ISS). The GEDI system is equipped with three identical 1064 nm wavelength lasers. Two of the laser beams are in full power, while the other one is split into two beams, thus generating in total four beams. Through dithering of each laser beam, GEDI acquires data along eight ground tracks. The GEDI tracks are 600 m across, and 60 m along track, with a footprint diameter of 25 m [48]. GEDI data are processed and made available by the Land Processes Distributed Active Archive Center (LP DAAC). First, the received waveforms are smoothed in order to reduce the noise in the signal. Waveform smoothing is performed by means of a Gaussian filter with various widths. As mentioned in the ATBD (Algorithm Theoretical Basis Document), currently a width of 6.5 ns was used for the Gaussian filter (smooth width). The processed data are issued by six configurations of algorithms (a1 to a6), representing different threshold and smoothing settings. In the case of water surfaces, the location of the water surface peak in the waveform is determined using a second Gaussian filtering. The width of the second Gaussian filter (Smoothwidth_Zcross) is fixed to either 3.5 or 6.5 ns.

Over our studied areas, there are between 3 and 13 acquisitions per lake during the first six months of available GEDI data (Table 3), corresponding to the time period between mid-April 2019 and end of October 2019. GEDI measures vertical structures using a 1064 nm laser pulse, and the echoed waveforms are digitized to a maximum of 1246 bins with a vertical resolution of 1 ns (15 cm). Over water surfaces, the recorded waveforms are unimodal (single peak), and similar in form to the transmitted pulse (i.e., the waveforms have a Gaussian like form). The estimation of water levels therefore relies on the accurate determination of this single peak within the waveform. In this study, L1B and L2A GEDI data products are used [49].

**Table 3.** Global Ecosystem Dynamics Investigation (GEDI) acquisition dates between April and October 2019 and available GEDI shot count over the studied lakes.

| Lake | Number of Shots | Dates |
|---|---|---|
| Geneva | 12,195 | 20/04; 04/05; 28/05; 20/06; 01/07; 04/07; 16/07; 18/07; 29/08; 02/09; 21/09; 23/09; 13/10 |
| Neuchâtel | 8522 | 21/04; 28/04; 29/05; 24/06; 20/07; 03/08; 18/08; 19/09; 28/09; 16/10; 25/10 |
| Thun | 1774 | 20/04; 21/04; 18/07; 18/08; 27/09; 25/10 |
| Lucerne | 1813 | 20/04; 22/05; 23/06; 28/06; 18/07; 05/09; 27/09 |
| Zug | 964 | 04/05; 17/06; 04/07; 13/07; 14/09; 23/09 |
| Zürich | 2433 | 02/05; 04/05; 11/05; 08/06; 04/07; 01/09; 23/09 |
| Walensee | 1083 | 20/04; 02/05; 23/06; 27/09; |
| Sempach | 239 | 04/05; 22/05; 08/06; 01/09 |
| Sarnersee | 203 | 20/04; 18/07; 23/07; |

### 2.2.3. In Situ Water Levels

Water level records from in situ gauge stations over Swiss lakes were made available by the Hydrology Department of the Federal Office for the Environment (FOEN) [50]. FOEN is currently monitoring the level and the quality of surface through a network composed of 260 gauge stations distributed across Switzerland. In this study, lake stages acquired with a temporal frequency of 10 mn from 12 stations over 10 lakes (see Figure 1 for their location and Table 4 for more details) were used to validate radar and lidar altimetry-based water levels.

**Table 4.** Information related to the in situ gauge stations used in this study: lake, station name and FOEN ID, geographical location, base level elevation, and geoid height from CHGeo2004 geoid model.

| Lake | Mean Area (km²) [51] | Station | Station ID | Longitude [1] (°) | Latitude [1] (°) | $h_0$ (m) | N (m) | Validation Period |
|---|---|---|---|---|---|---|---|---|
| Geneva | 580 | St-Prex | 2027 | 6.4611379 | 46.4827956 | 421.92 | 49.93 | 01/1995–01/2020 |
| Geneva | 580 | Sécheron | 2028 | 6.1523679 | 46.218622 | 421.82 | 49.83 | 01/1995–01/2020 |
| Neuchâtel | 215 | Grandson | 2154 | 6.6423606 | 46.8057745 | 478.82 | 49.86 | 01/1995–01/2020 |
| Thun | 48 | Spiez, Kraftwerk BKW | 2093 | 7.6645587 | 46.6967403 | 608.09 | 50.09 | 09/2018–01/2020 |
| Lucerne | 114 | Lucerne | 2207 | 8.3198266 | 47.0548813 | 482.03 | 48.08 | 01/1995–01/2020 |
| Lucerne | 114 | Brunnen | 2025 | 8.6037922 | 46.9934707 | 482.49 | 48.52 | 01/1995–01/2020 |
| Zug | 38 | Zug | 2017 | 8.5143326 | 47.1678740 | 451.62 | 47.59 | 01/2018–01/2020 |
| Zürich | 68 | Zürich | 2209 | 8.5504684 | 47.3547707 | 453.21 | 47.32 | 02/2016–01/2020 |
| Zürich (Obersee) | 20 | Schmerikon | 2014 | 8.9400917 | 47.2247744 | 457.25 | 47.32 | 04/2018–01/2020 |
| Walensee | 24 | Murg | 2118 | 9.2100321 | 47.1133767 | 467.07 | 48.13 | 01/1995–01/2020 |
| Sempach | 14 | Sempach | 2168 | 8.1891830 | 47.1342944 | 551.97 | 48.03 | 09/2018–01/2020 |
| Sarnen | 8 | Sarnen | 2088 | 8.2424260 | 46.8877257 | 520.36 | 49.37 | 09/2018–01/2020 |

[1] Geographical coordinates were obtained using the NAVREF tool from SwissTopo [52].

### 2.2.4. CHGeo2004 Geoid Model

The national geoid model of Switzerland CHGeo2004 was obtained by combining data from three different methods: gravity, vertical deflections, and GPS/leveling. Its accuracy was found to be about 2–3 cm through comparisons against independent data [53]. The CHGeo2004 geoid model is made available by the Federal Office of Topography as a grid of 1 km of spatial resolution at [54].

### 2.3. Altimetry-Based Water Levels

#### 2.3.1. Measurement Principle

Radar altimetry is a technique allowing to retrieve the surface height based on the difference between the altitude of the satellite (H) and the distance between the satellite and the surface or range ($R_0$). The satellite altitude is accurately estimated at centimeter level using precise orbit determination techniques and the range is derived from the two-way travel time of the electromagnetic wave emitted by the sensor ($\Delta t$) considering a velocity equals to the speed of light in the vacuum (c) [55,56]:

$$R_0 = c\Delta t/2, \tag{1}$$

Several corrections to the range are applied to take into account propagation delays due to the presence of the atmosphere and geophysical effects [55,56]. Over inland water bodies, the surface height is given by [57]:

$$h = H - (R_0 + \Delta R_{iono} + \Delta R_{dry} + \Delta R_{iono} + \Delta R_{solid\ Earth} + \Delta R_{pole}) - N, \tag{2}$$

where $\Delta R_{iono}$, $\Delta R_{dry}$, $\Delta R_{wet}$, $\Delta R_{solid\ Earth}$, and $\Delta R_{pole}$ are the atmospheric refraction range correction due to the free electron content associated with the dielectric properties of the ionosphere, the atmospheric refraction range corrections due to the dry gas component and the water vapor and the cloud liquid water content of the troposphere, the corrections accounting for crustal vertical motions due to the solid Earth and pole tides, respectively, and N is the geoid height.

For land hydrology, $R_0$ values are obtained through processing the radar altimeter echoes or waveforms with the Offset Center of Gravity (OCOG) retracking algorithm [58] following [59]. Ionosphere and wet troposphere corrections are derived from total electron content and atmosphere models respectively as it is usually done over land [57].

### 2.3.2. Generating the Time-Series of Water Levels from Radar Altimetry Data Using AlTiS

Altimetry data were manually processed using the Altimetry Time Series (AlTiS) software recently developed by the Centre de Topographie des Océans et de l'Hydrosphère (CTOH), an observation service labelled by Institut National des Sciences de l'Univers (INSU) based at Laboratoire des Etudes en Géophysique et Océanography Spatiales (LEGOS), dedicated to the production and distribution of data and services related to radar altimetry. In this framework, AlTiS was developed to replace and enhance the Multi-mission Altimetry Processing Software (MAPS), jointly developed by CTOH and Environnements et Paléo-Environnements Océaniques et Continentaux (EPOC) [60,61]. AlTiS is an Open Source project under CeCill license. AlTiS is a Python-based Graphical User Interface (GUI) using wxWidgets cross platform library which allows to:

- Read radar altimetry data from ERS-2, ENVISAT, JASON-1/2/3, SARAL, and SENTINEL-3A and 3B radar altimetry missions.
- Display the different variables contained in the Geophysical Data Records (GDR) of each mission including H, $R_0$, the different corrections applied to $R_0$, h, automatically computed from (2) when reading the data, as well as several other variables such as the backscattering coefficients and the pulse peakiness [62] at the different microwave frequencies, the brightness temperatures at the different frequencies measured by the radiometer on-board the satellite platform, and the normalized index defined by CTOH to help for the statistical analysis [63], with Landsat True color image supplied by the Global Imagery Browse Services (GIBS from NASA's Earth observations [64]) as background.
- Manually select the valid data/remove the invalid data contouring them using the mouse.
- Generating the time series of water levels computing the median and mean values and the associated median absolute deviation and standard deviation for each cycle. Note that the different altimeter tracks are processed individually. In this study, median values and associated median absolute deviations computed each cycle are used to minimize the potential impact of residual outliers on small number of observations due to the moderate width of the lakes under the altimeter tracks (see Table 5).

**Table 5.** Information on the radar altimetry crossings over the Swiss lakes considered in this study.

| Lake | Altimetry Missions | Track | Repeat Orbit (Days) | Distance over the Lake (km) | Valid Data |
|---|---|---|---|---|---|
| Geneva | ERS-2/ENVISAT/SARAL | 0846 | 35 | 9.4 | Yes/No/Yes |
| Geneva | ERS-2/ENVISAT/SARAL | 0887 | 35 | 5.6 | Few/Yes/Yes |
| Geneva | JASON-1/2/3 | 044 | 10 | 5.6 | No/Yes/Yes |
| Geneva | SENTINEL-3A | 0358 | 27 | 12.2 | Yes |
| Geneva | SENTINEL-3A | 0741 | 27 | 4.7 | Yes |
| Geneva | SENTINEL-3B | 0741 | 27 | 11.2 | Yes |
| Neuchâtel | ERS-2/ENVISAT/SARAL | 0343 | 35 | 5.6 | Yes/No/No |
| Neuchâtel | ERS-2/ENVISAT/SARAL | 0846 | 35 | 12.9 | No/Few/No |
| Neuchâtel | SENTINEL-3A | 0358 | 27 | 3.9 | Yes |
| Neuchâtel | SENTINEL-3A | 0741 | 27 | 6.1 | Yes |
| Thun | SENTINEL-3A | 0085 | 27 | 5.5 | Yes |
| Lucerne | ERS-2/ENVISAT/SARAL | 0257 | 35 | 2.4/3.3/2.4 [1] | Few/Few/Few |
| Lucerne | ERS-2/ENVISAT/SARAL | 0760 | 35 | 5.8 | Yes/Yes/Yes |
| Lucerne | SENTINEL-3A | 0199 | 27 | 1.3/0.8 [1] | Yes |
| Lucerne | SENTINEL-3A | 0586 | 27 | 8.00 | Yes |
| Zürichsee | SENTINEL-3A | 0586 | 27 | 3.1 | Yes |
| Obersee (Zürich) | SENTINEL-3B | 0313 | 27 | 1.5 | Yes |
| Walensee | ERS-2/ENVISAT/SARAL | 0760 | 35 | 1.5 | Few/No/No |
| Walensee | SENTINEL-3B | 0700 | 27 | 1.1 | Yes |

[1] Due to the shape of the lake, ERS-2/ENVISAT/SARAL (SENTINEL-3A) ground-track 0257 (0199) crosses Lucerne Lake in three (two) different locations.

AlTiS can be downloaded at [65] and altimetry data used by AlTiS can be requested to CTOH at [66].

### 2.3.3. Generating the Time-Series of Water Levels from ICESat-2 Lidar Data

The water heights are provided by NSIDC including all corrections (atmospheric and geophysical). For each pass selected the ATL13 data products were used. They are composed of precise latitude, longitude, and elevation for every received photon arranged by beam in the along-track direction, see [67]. We applied geoid corrections for each individual lake height estimates, using EGM2008 model [68]. For large lakes it has been shown that the global model may have errors up to several decimeters [69], but over the Swiss lakes of this study, errors introduced in the time series due to the geoid model can be considered as negligible, except for the Lake Geneva, were significant part of the errors can be attributed to geoid model errors. All measurements (after geoid corrections are applied) along a pass are then averaged in order to produce lake height time series that can then be compared to in situ data. The average calculation is an iterative data processing including outliers detection (and removal) based on standard deviation of the individuals measurements.

### 2.3.4. Generating the Time-Series of Water Levels from GEDI Lidar Data

Over water, unimodal waveforms are observed in general and only two different sets of algorithms produce different elevations. Sets a1 and a4 are similar, and sets a2, a3, a5, and a6 are similar. Fayad et al. [70] showed that the parameters used in algorithm a1 (Smoothwidth_zcross of 6.5 ns) provide more precise elevations in comparison to algorithm a2 (Smoothwidth_zcross of 3.5 ns). Therefore, in this study, only the elevations produced from algorithms a1 were analyzed. As such, we extracted for each GEDI shot from the L2A data product the latitude and longitude, as well as the elevation of the lowest peak or mode (for water surfaces only one peak is available).

Not all GEDI acquisitions are viable, as atmospheric conditions and clouds can affect them. Here, two filters were applied to remove unusable returns. The first filter applied removes waveforms with reported elevations that are significantly higher than the corresponding Shuttle Radar Topography Mission (SRTM) DEM elevation [71] (i.e., we removed all waveforms where |GEDI elevation—SRTM| > 100 m). Since we are only interested with waveforms that are acquired over water, we removed all waveforms having two or more peaks or modes; 29,226 GEDI shots out of 67,005 available shots were exploitable and analyzed over the eight studied lakes (43.6%) (Table 1).

Then, valid data used to estimate the GEDI-based water levels for each beam are selected upon the following criteria: (i) the amplitude of the smoothed waveform's lowest detected mode (zcross_amp) is lower than 3700 amplitude counts (AC), and (ii) the height value is encompassed between the average height per date plus or minus the corresponding standard deviation.

### 2.3.5. Levelling of the Different Water Level Datasets

Given that gauge station elevations are provided with respect to the Swiss height measurement system (LN02), and the water levels derived from AlTiS are given with reference to EGM2008 geoid model [68], and the geolocated GEDI footprints are relative to the WGS84 ellipsoid, a conversion of the different altimetry datasets is necessary for a consistent analysis. Two steps are necessary to perform a conversion between WGS84 and LN02 elevations. First, ellipsoidal elevations of GEDI footprints are converted to orthometric elevations with respect to the new Swiss height system LHN95 (Landeshöhennetz 1995) using the following equation:

$$H_{LNH95} = h_{wgs84} - N_{CHGEO2004} \tag{3}$$

where $H_{LHN95}$ is the converted altimetry elevation with respect to LHN05, and $N_{CHGEO2004}$ the Swiss gravimetric geoid heights. Altimetry elevations, which are now orthometric

elevations with respect to LHN95, are finally converted to the Swiss height system (LN02) by means of three grids since height differences between LHN95 and LN02 cannot be modeled by a single offset. This is due to their different way of gravity reduction, the treatment of vertical movements, and the constraints introduced in LN02. The conversion between orthometric LHN95 heights and LN02 heights could be made using the following equation [72]:

$$H_{ln02} = H_{LNH95} + H_{norm} - H_{scale} - \frac{\Delta g_{boug}}{g} H_{LHN95} \tag{4}$$

where $H_{ln02}$ are the altimetry elevations with respect to the Swiss height system (LN02), $H_{norm}$ is a 1 km grid describing the difference between LN02 and normal heights, $H_{scale}$ is a 1 km grid scale factor used to transform between normal heights and orthometric heights, $D_{gboug}$ is a 1 km grid representing the Bouguer anomalies, and g is the average normal gravity equal to 980,000 mGal. The Swiss geoid grid (CHGeo2004), as well as the three grids used in the transformation between LHN95 and LN02 heights, were obtained from the Swiss Federal Office of Topography [73].

2.3.6. Validation of Altimetry-Based Lake Water Levels

The lake water levels derived from radar and lidar altimetry were validated through comparisons against in situ gauge records made available by FOEN. All data were converted to the same reference. To limit systematic error due to the distance between in situ and satellite measurements, the more accurate geoid model available over the area (i.e., CHGeo2004) was chosen as a common reference.

Bias (computed as the difference between the in situ and altimetry-based water stages), root mean square error (RMSE), relative RMSE defined as the ratio between the RMSE and the difference between the maximum and minimum value of the in situ data during the observation period, and Pearson correlation coefficient (R) were estimated between in situ and satellite measurements. As GEDI is acquiring data with a different incidence angles, all these statistical measures are computed for every beam. For some overpasses, due to the configuration of the acquisition, not every beam encompasses the lakes surface in their footprint resulting in different sampling of the lakes surface for each beam. For GEDI, as the number of available dates of acquisition is quite limited and does not allow to compute RMSE and R for the time-series of water levels (e.g., when number of dates is below 10), bias and RMSE are computed for every date when the number of acquired valid GEDI shots is higher than 10.

## 3. Results

Altimetry time-series of water levels were generated over the following Swiss lakes: Geneva, Neufchâtel, Thunersee, Vierwaldsee, Zurichsee, Walensee, and Zugersee using data from ERS-2, ENVISAT, JASON-1/2/3, SARAL, and SENTINEL-3A and B. Due to the coarse coverage of current altimetry missions on the Earth surface, not every mission ground-tracks cross every lake. However, the lakes chosen in this study are all covered by a SENTINEL-3 ground-track. Information on the radar altimetry crossings over the Swiss lakes considered in this study are presented in Table 5. In contrast, GEDI ground-tracks crossed several times each lake between 20 April 2019 and 25 October 2019. Note that when a lake has several in situ stations (Saint Prex and Sécheron for Lake Geneva; Lucerne and Brunnen for Lake Lucerne in this study), the results for only one of them are presented. The results obtained for the second one are almost identical.

*3.1. Validation of Radar Altimetry-Based Water Levels*

When considering radar altimetry data prior to the ones acquired by JASON-3 and SENTINEL-3 A and B missions, a lot of lake crossings do not provide any valid data or too few to build continuous time-series of water levels over the Swiss lakes. On the contrary, for all SENTINEL-3 crossings, time-series of water levels were derived, as well as for the only Jason-3 crossing over the Geneva Lake (Table 3). For all the altimetry-based time-series

of water levels, comparisons were performed against in situ data. The statistical results are reported in Figures 2 and 3.

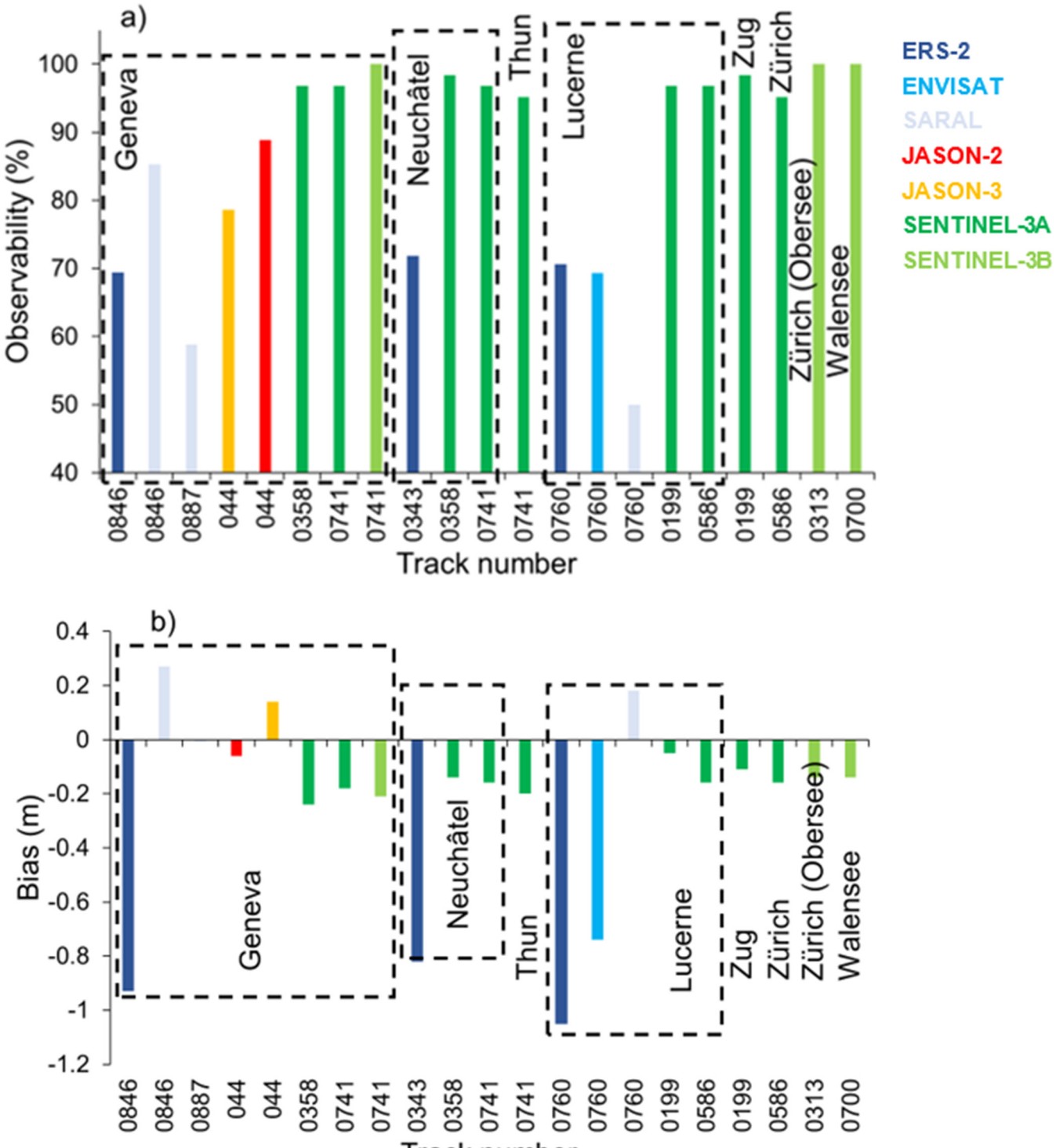

**Figure 2.** Results of the comparison of the water levels measured in situ and derived from RA observations over 8 Swiss lakes: (**a**) Observability of the lake by RA missions (%), (**b**) Bias (m). Dark blue, light blue, pale blue, red, orange, green and light green bars correspond to ERS-2, ENVISAT, SARAL, JASON-2, JASON-3, SENTINEL-3A, and SENTINEL-3B, respectively.

A wide range of cross-section lengths are considered in this study, ranging from 0.8 to 12.9 km (Table 5). Due to the orbit characteristics of the 10-day repeat orbit missions,

only Geneva Lake is crossed by one ground-track from the JASON-1/2/3 missions. If no valid observations from JASON-1 are available, it is frequently sampled by JASON-2 and -3 with observability greater than 78 and 88% respectively (Figure 2a). Half of the lakes are crossed by one or two ground-tracks from the missions with a 35-day repeat orbit (Table 5). On the seven crossings, valid data were found for 3, 1, and 3 of them using ERS-2, ENVISAT, and SARAL, respectively. In these cases, the lake surface was observed ~70% of the time with ERS-2 and ENVISAT missions whereas using SARAL observations, this percentage ranges from 50% for Lucerne Lake, to ~60% and even over 85% over Geneva Lake (Figure 2a). All the eight lakes are observed with the missions on the 27-day repeat orbit, with 9(3) crossings over 5(3) lakes for SENTINEL-3A(3B), respectively (Table 4). An excellent observability of the lakes is reached with these missions (>95% and even equals to 100% for SENTINEL-3B). Considering the bias, large negative values (i.e., the water levels measured by the RA missions are lower than the ones measured in situ, meaning that the altimeter range estimate is too long) were obtained for the missions on the 35-day repeat orbit, ranging from 0.27 to −1.05 m. A lower deviation is obtained with ERS-2 than with SARAL with values ranging from −1.05 to −0.82 m and from −0.01 to −0.27 m, respectively (Figure 2b). Low biases are estimated using the missions on the 10-day repeat orbit: −0.06 and 0.14 m for JASON-2 and JASON-3, respectively (Figure 2b). For the mission on the 27-day repeat orbit (SENTINEL-3A and B), a very stable bias is determined: $(-0.17 \pm 0.04)$ m.

Examining the ability of RA missions to monitor temporal variations of water levels, RMSE and correlation coefficients between in situ and altimetry-based water levels were determined. They are presented in Figure 3. Large RMSE values, ranging from 0.28 to 0.54 m, are observed for ERS-2 and ENVISAT (Figure 3a). For these missions, R ranges from very low (0.22 for ENVISAT on Lake Lucerne) to rather good (0.63 for ERS-2 on lakes Geneva and Lucerne) (Figure 3b). Lower RMSE (0.21 and 0.13 m) and higher or equal R values (0.61 and 0.75) were obtained for JASON-2 and JASON-3, respectively. Low RMSE (values lower than 0.1 m except for SENTINEL-3A ground-track 0199 over Lake Lucerne) and high R (values higher than 0.85 except for SENTINEL-3A ground-tracks 0199 and 0586 over Lake Lucerne) were found (Figure 3).

### 3.2. Validation of Lidar Altimetry-Based Water Levels

3.2.1. Validation of ICESat-2-Based Lake Water Levels

The observability of the six Swiss lakes under the ICESat-2 ground-tracks ranges between 45 and 100% (Figure 4a). An almost constant bias was found over the six Swiss lakes with an average value of $(0.42 \pm 0.03)$ m (Figure 4b). In spite of a quite a low number of observations, ranging between 3 and 12 depending on the considered lake, RMSE and R were found below (over) 0.06 m (0.95), respectively (Figure 5a,b).

3.2.2. Validation of GEDI-Based Lake Water Levels

The number of cross-sections between GEDI ground-tracks and the Swiss lakes considered in this study is much smaller than the ones from the RA missions as it ranges between 3 over Sarner Lake and 13 over Lake Geneva. As GEDI mission collects data on a non-repetitive orbit using eight different beams, the observability is here defined as the number of times each beam acquires valid data over a lake based on the criteria defined in Section 2.3.4, following [70], divided by the number of times the GEDI ground-tracks cross the lake. The observability is logically a function of the lake area: observability is higher over the large lakes, generally over 60% over Lakes Leman, Neuchâtel, Lucerne, and Zürich (including Obersee), and lower (below 50%) over the small lakes (Sempach and Sarnen) for all the beams. On lakes of intermediate size, between 20 and 50 km$^2$ of area, the observability is highly variable among the beams (Figure 6a).

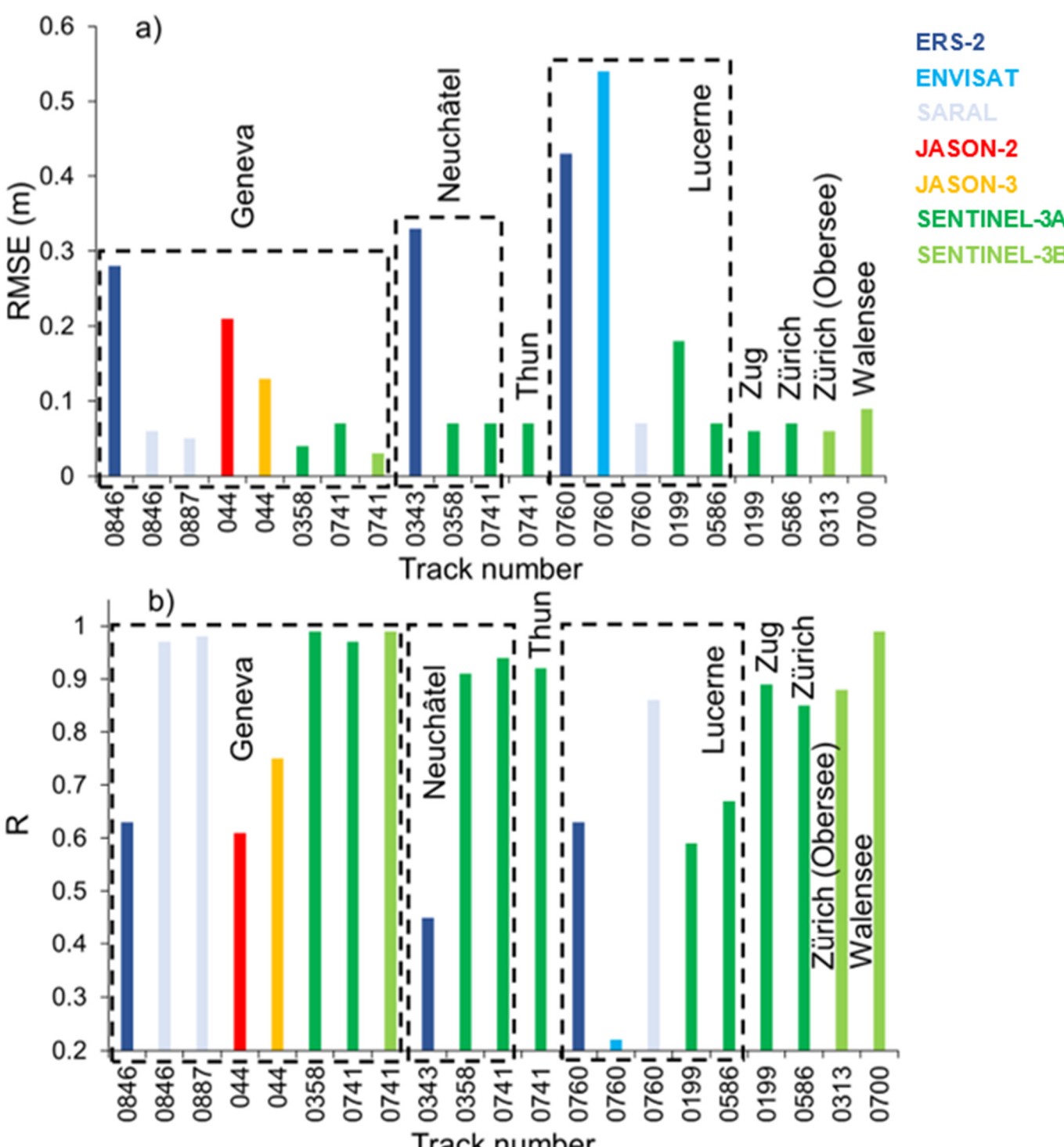

**Figure 3.** Results of the comparison of the water levels measured in situ and derived from RA observations over 8 Swiss lakes: (**a**) RMSE (m), (**b**) R. Dark blue, light blue, pale blue, red, orange, green and light green bars correspond to ERS-2, ENVISAT, SARAL, JASON-2, JASON-3, SENTINEL-3A, and SENTINEL-3B, respectively.

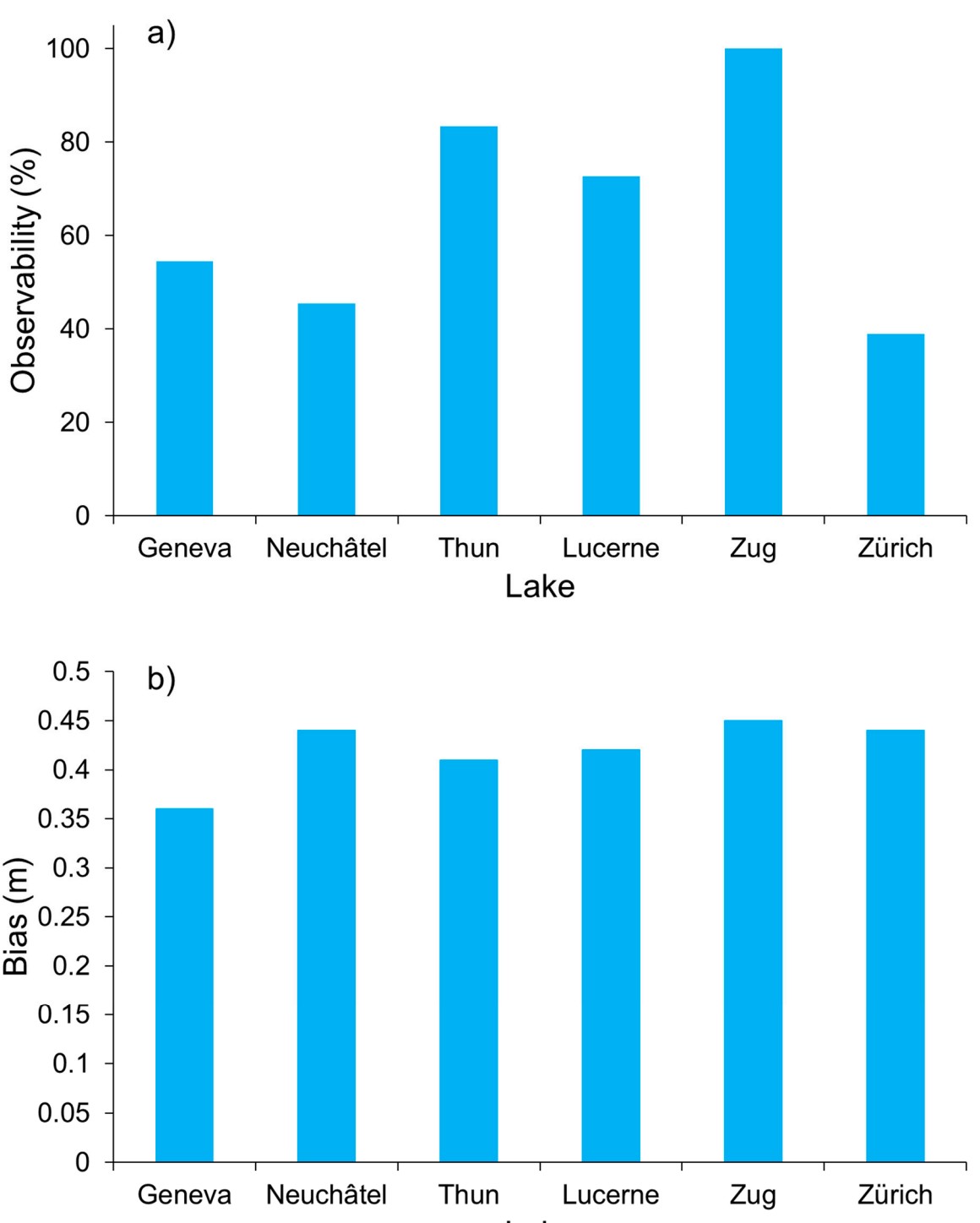

**Figure 4.** Results of the comparison of the water levels measured in situ and derived from ICESat-2 observations over 6 Swiss lakes: (**a**) Observability of the lake by ICESat-2 (%), (**b**) Bias (estimated—in situ) (m).

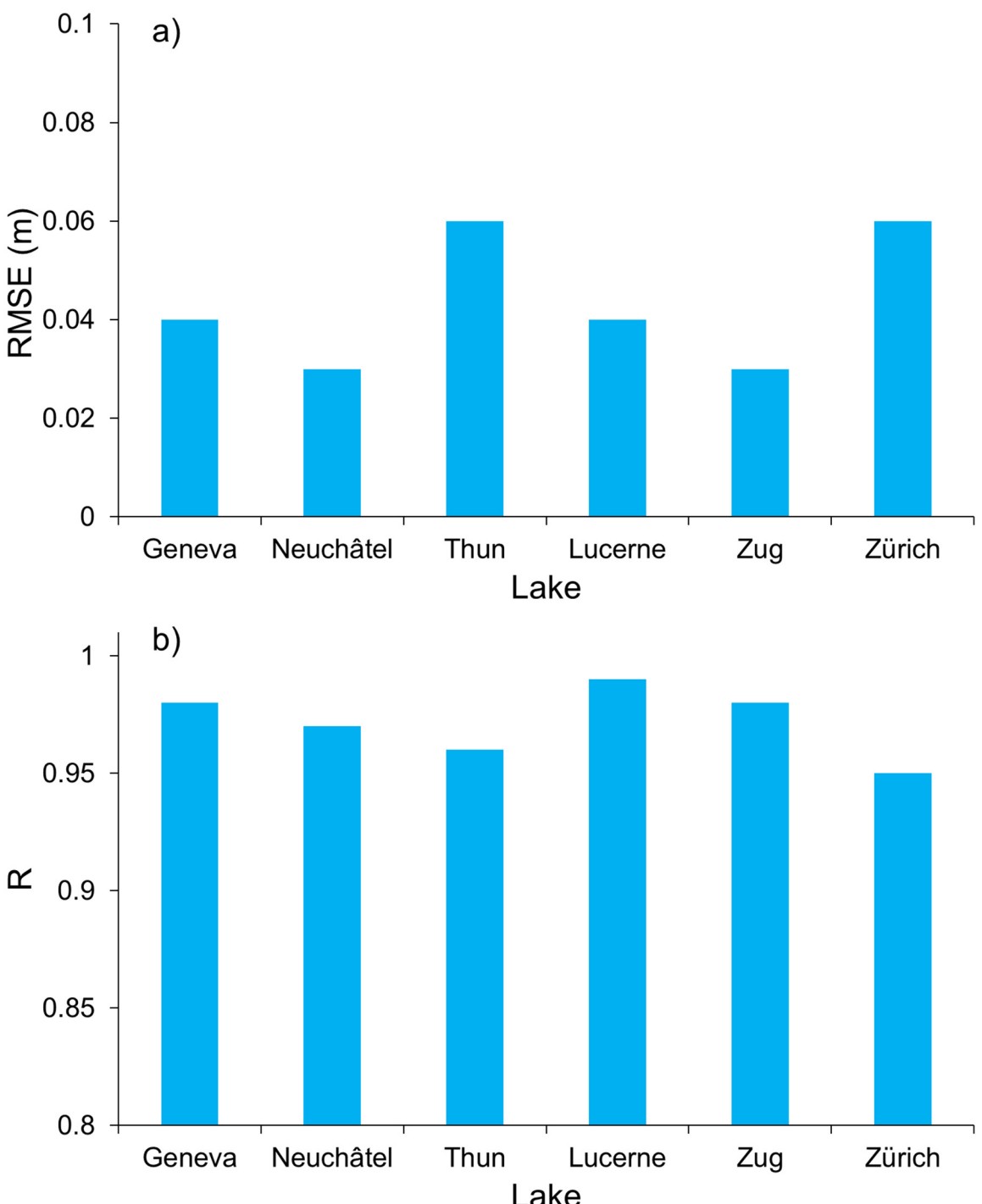

**Figure 5.** Results of the comparison of the water levels measured in situ and derived from ICEsat-2 observations over 6 Swiss lakes: (**a**) RMSE (m), (**b**) R.

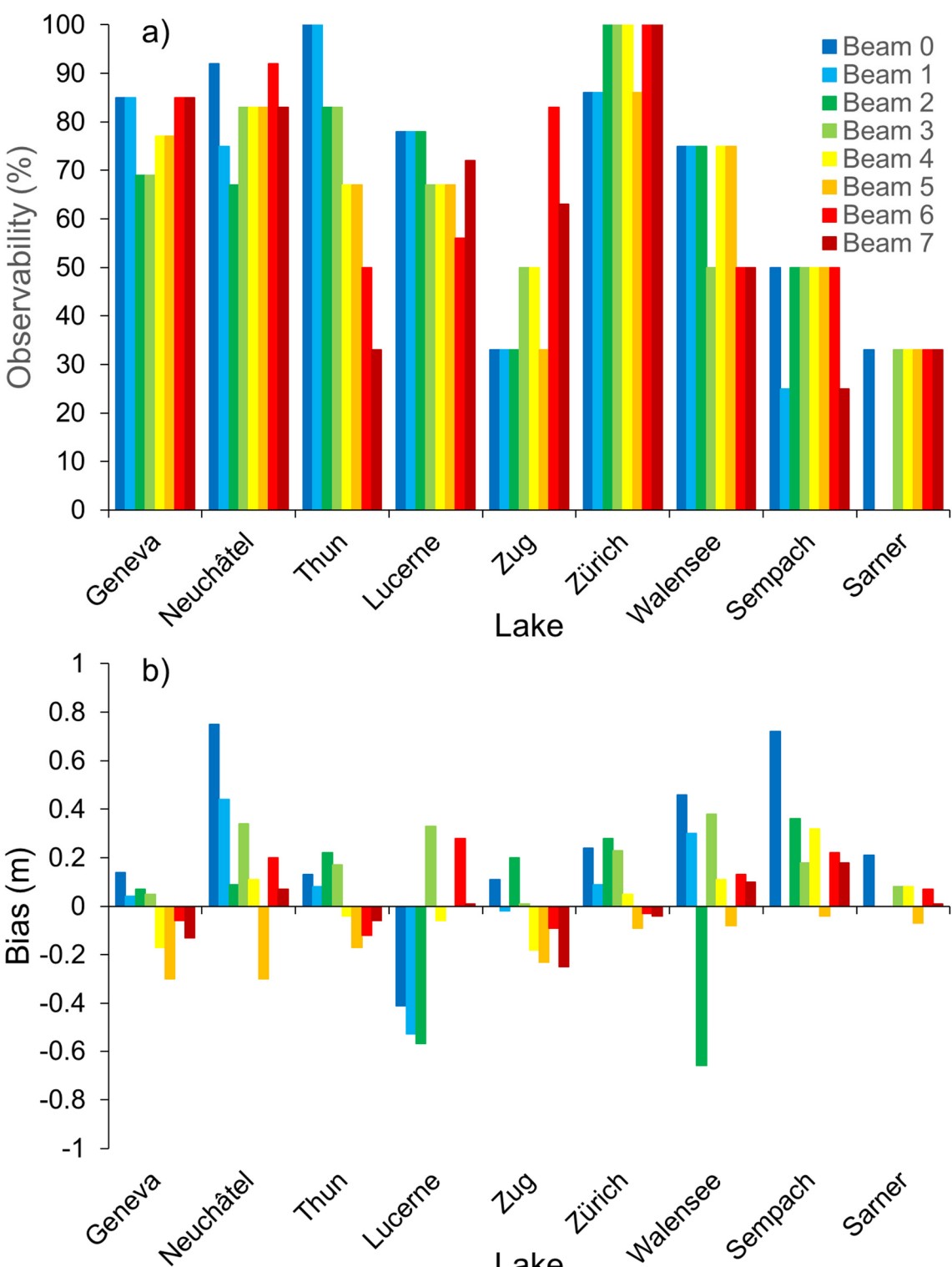

**Figure 6.** Results of the comparison of the water levels measured in situ and derived from GEDI observations over 9 Swiss lakes: (**a**) Observability of the lake by GEDI (%), (**b**) Bias (estimated—in situ) (m). These statistics were computed for each of the 8 GEDI beams.

As the number of GEDI overpasses is below 10 except over Lakes Geneva and Neuchâtel, RMSE and R were only determined over these two lakes, but the global biases (estimated for the whole observation period) were estimated over every lake and every beam. Bias estimates are presented in Figure 6b. These values exhibit a wide range of values, from −0.37 to 0.62 m depending on the beam and the lake considered (Figure 4b). The largest values in modulus are most of the times obtained for beams 0 to 5. Figure 7 summarizes the results obtained for all the lakes by each beam. Higher bias values, ranging, in modulus, between 0.13 and 0.21 m, are obtained for beams 0, 3, and 5. High variability of the bias (std > 0.2 m) can be observed for beams 0, 1, 2, and 4. Only beams 6 and 7 exhibit low mean bias and associated std.

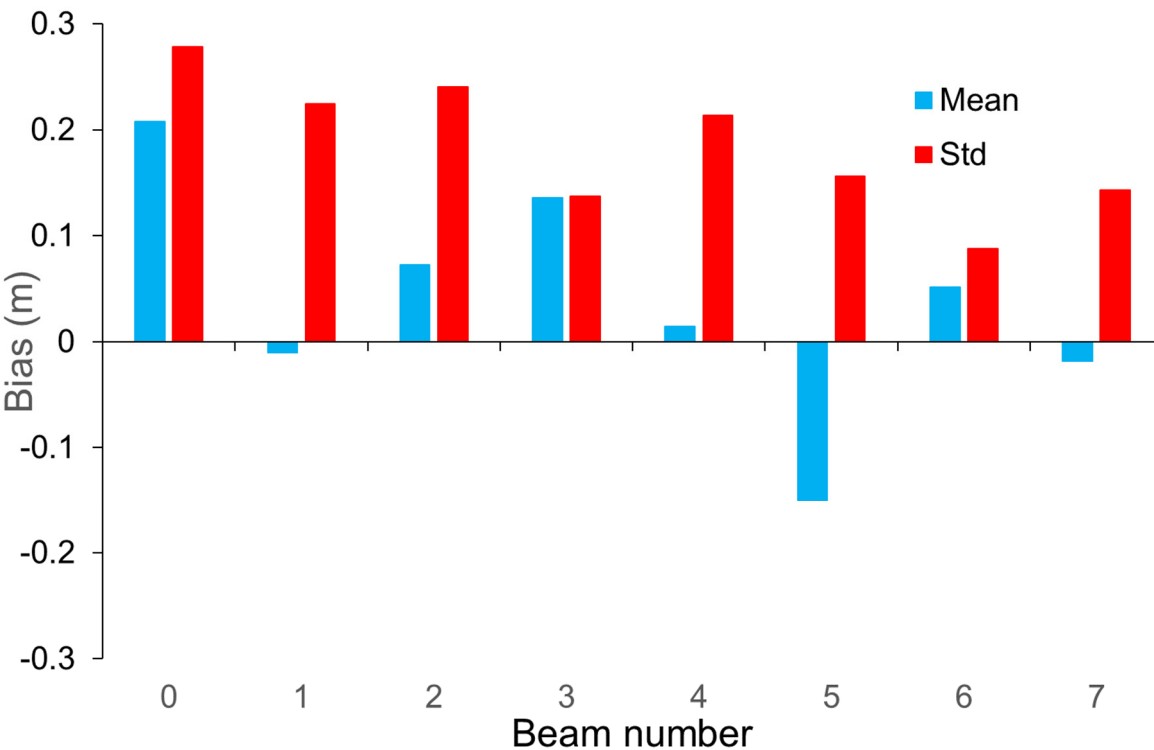

**Figure 7.** Mean bias (m), in blue, and associated std (m), in red, over the nine Swiss lakes, by GEDI beam.

RMSE and R values were estimated for all beams over Lakes Geneva and Neuchâtel as the number of data commonly available at the same date between GEDI and the in situ gauge record ranges is between 9 and 11 for Lake Geneva and between 7 and 10 for Lake Neuchâtel (Figure 8). For Lake Geneva, R values higher than 0.75 (with *p*-values lower than 0.06) were found for beams 0, 1, 6, and 7 with corresponding RMSE ranging from 0.16 (beam 2) to 0.30 (beam 7). For the other beams, similar values of RMSE can be observed but with very low and unsignificant correlations (R values below 0.25 in modulus and *p*-values above 0.6). For Lake Neuchâtel, RMSE are generally higher (up to 0.51 m) and all are R values are negative.

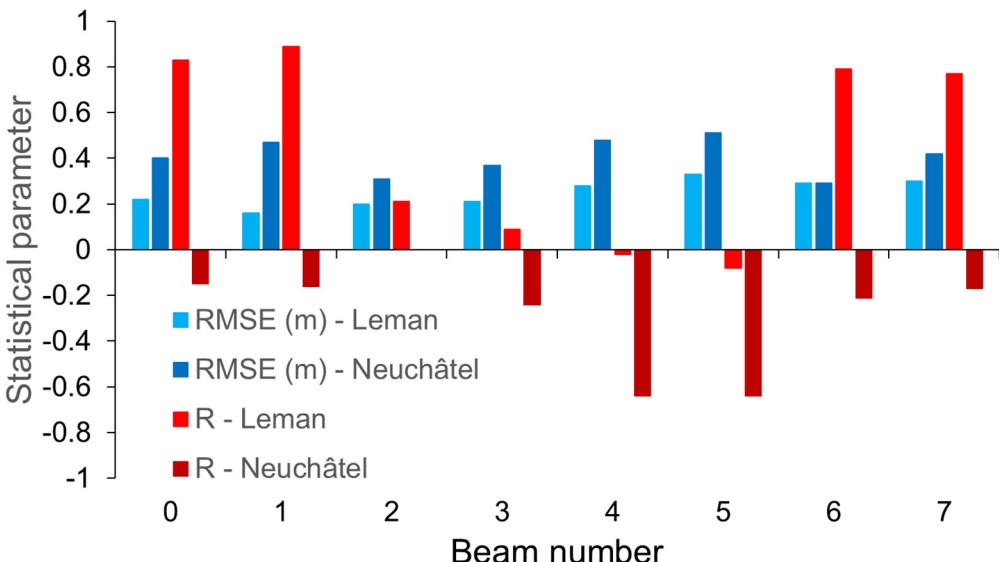

**Figure 8.** RMSE (m), in blue, and R (m), in red, over the Lakes Leman and Neuchâtel, by GEDI beam.

## 4. Discussion

### 4.1. Availability and Accuracy of Radar Altimetry-Based Water Levels in Mountainous Areas

Since the beginning of the high accuracy RA era starting with the launch of ERS-1 and Topex/Poseidon in 1991 and 1992, respectively, RA data were widely used to provide a long-term monitoring of lake levels all over the world with a good accuracy (e.g., [9,10,57]). The main factors responsible for a non-acquisition of data or a decrease in accuracy of the altimetry measurement are the following: (i) the presence of topography in the surroundings of the waterbody causing a loss of the satellite tracking [74], (ii) inhomogeneities in the altimeter footprint due to the small size of the waterbody [16,17], or the presence of ice on the surface of the waterbody at high latitudes [75–77].

In the case of the Swiss lakes considered in this study, as the cross-section between the RA ground-tracks and the lake's surface is greater than 1 km, and most of the times, the cross-section is several km in length, a very good accuracy could have been expected for all the missions. Considering the results obtained, it clearly appears the presence of topography surrounding the Swiss lakes considered in this study is responsible for either the lack of valid data or their low availability (<30% of observability). For the missions no-operating in OL or DEM tracking-mode [13,14], it was not possible to retrieve water levels from 4, 6, and 4 out of 7 crossings for ERS-2, ENVISAT, and SARAL data, respectively. In the other cases, the radar altimeter either remained locked on the top of the surrounding hills or received no echo during the reception phase and switched to pursuit mode for identifying new echoes. This is clearly illustrated in Figure 7 which presents along-track profiles from ERS-2, ENVISAT, and SARAL over Neuchâtel Lake. The lake level is around 430 m. Reliable data over the Neuchâtel Lake were found for four cycles for ERS-2 (Figure 9a), three cycles for ENVISAT (Figure 9b), and one for SARAL (Figure 9c). For SARAL, most of the data are acquired on the surrounding areas, a couple of hundred meters above the lake level. Note that three times more virtual stations (VS) were constructed with the measurements from ERS-2 (1995–2002 on its nominal orbit) reprocessed by CTOH [40] than from ENVISAT (2002–2010 on its nominal orbit) and an equal number as SARAL (2013–2016 on its nominal orbit). Not taking into account the bias, low RMSE and high R values are obtained using SARAL data owing to the use of the Ka-band [11]. For such large waterbodies, higher RMSE and lower R values than usual are obtained using ERS-2 and ENVISAT (Figure 3). As the temporal variations of the lake surfaces (~1 m annually), the RMSE values obtained for these two missions represent a large RRMSE (Figure A1) which can account for the low R values. The only crossing between JASON-1/2/3 over the studied lakes provides valid time-series for JASON-2 and 3. If the results are quite accurate,

the RMSE is a little bit high (>10 cm), and the correlation of only 0.61 and 0.75 for JASON-2 and 3, respectively (Figure 3). These somehow lower than expected results are likely to be due to the locations of the 10-day repeat period ground-track 044 on the eastern bank of the lake where the altimeter footprint encompasses the cities of Villeneuve, Montreux, and Vevey as well as some mountainous areas. The higher observability of JASON-3 compared to JASON-2 (88.8% against 78.5% in Figure 2a) and its better statistical results when comparing to in situ gauge records mentioned above (Figure 3) can be attributed to the OL or DEM tracking mode as in [13].

The results obtained using data from the missions operating in LRM acquisition mode are outperformed by the ones obtained by SENTINEL-3 operating in SAR acquisition mode and OL or DEM tracking mode. The observability is greater than 95% in the case of the nine cross-sections between a lake and a SENTINEL-3A ground-track and equals to 100% in the case of the three cross-sections between a lake and a SENTINEL-3B ground-track (Figure 2a). The bias is almost constant ($-0.17 \pm 0.04$) m over the 12 VS. RMSE is lower or equal to 0.07 m and R higher than 0.85 when the cross-section between the ground-track and the lake surfaces is longer than 1 km (Figure 3 and Table 5). It is, for instance, the case for SENTINEL-3A ground-track 0741 crossing Geneva Lake on a shorter distance than JASON ground-track 044 and closer to the shore. In spite of this drawback, an RMSE of 0.07 m and a R of 0.97 were found, better than the ones obtained with JASON-2 and 3, illustrating the strong importance of the OL or DEM tracking and the SAR acquisition modes for the performances of RA missions. Lower performances were only found over the Lake Lucerne where the irregular shape of the lake likely impacts the radar echoes due to land contamination in the RA waveform.

The good results obtained in this study confirms the previous results concerning the retrieval of inland water levels using SENTINEL-3A (e.g., [14,24,61,76,78]) and shows that SENTINEL-3B performs as well as SENTINEL-3A even for inland water bodies surrounded by strong topography. Similar biases were found for the two missions even in the case of higher RMSE and lower R values. The data of the two satellites can be combined without considering any correction due to an inter-mission bias. For lakes crossed by several SENTINEL-3 ground-tracks, a resulting time-series of similar quality but with a higher temporal sampling can be obtained combining the information acquired on the different tracks. For Lake Geneva crossed by two SENTINEL-3A ground-tracks and one SENTINEL-3B one, the resulting bias, RMSE and R values are equal to $-0.21$ m, 0.06 m, and 0.96, respectively (Figure 10).

*4.2. Availability and Accuracy of Lidar Altimetry-Based Water Levels in Mountainous Areas*

Contrary to what could be expected, the observability is lower on larger lakes—39, 45, and 54% over Zürich, Neuchâtel, and Geneva (including Obersee) lakes (Figure 4a) whose areas are 88, 215, and 580 km$^2$ (Table 4), respectively—than over smaller lakes—100 and 83% over Zug and Thun lakes (Figure 4a) whose areas are 38 and 48 km$^2$ (Table 4), respectively. A good observability (72%) is found over Lake Lucerne (114 km$^2$) meaning that the most important factor for availability is the weather conditions. The stability of the bias over six different lakes (the std on the bias is equal to 0.03 m) and the very good performances for the retrieving of water levels with RMSE (Figure 5a), RRMSE (Figure A2), and R (Figure 5b) values similar to the ones obtained using SENTINEL-3 make ICESat-2 a very good candidate for densifying existing time-series of water levels from RA, creating new ones over water bodies not under current RA missions ground-tracks, levelling gauge stations in a close vicinity of ICESat-2 ground-tracks, and being used for the cal/val of the current and future RA missions, including the NASA/CNES Surface Water and Ocean Topography (SWOT) mission.

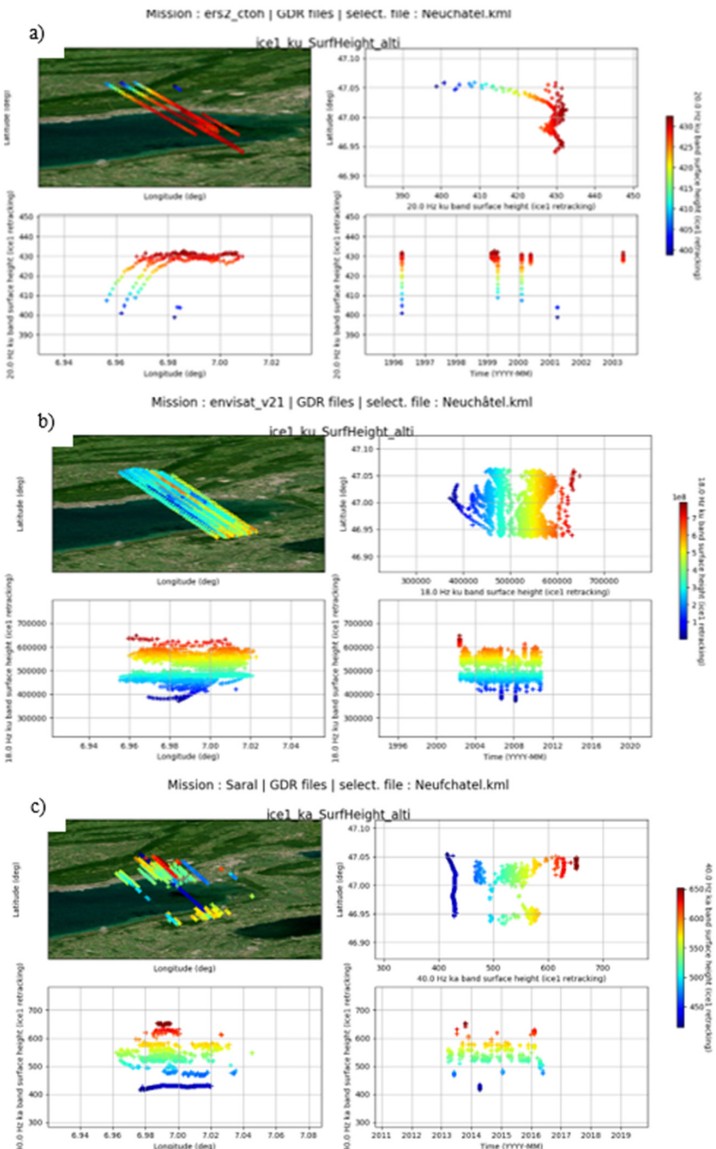

**Figure 9.** Along-track elevation profiles along the track 0343 of the 35-day repeat orbit: (**a**) ERS-2, (**b**) ENVISAT, (**c**) SARAL, taken from screenshots of the AlTiS software GUI. The altimeter height are provided in (m) in (**a**) and (**c**) an in (mm) in (**b**) respecting the units provided in the GDR files. Note that in the new version of the ENVISAT GDR (v3), they are provided in m contrary to the current v2.1. It will be possible to read and process this new dataset in the new version of AlTiS (1.9) available soon.

The overall observability using GEDI is quite low over most of the lakes considered in this study. Many of the acquisition dates over the Swiss lakes occurred in spring and fall when the presence of cloud is frequent over western Europe. Without applying thresholds on the zcross_amp parameter (see Section 2.2.3), the observability increases at the expense of the quality of the results with a rise of the bias variability compared to Figure 7, and a rise/reduction of the RMSE/R, respectively compared to Figure 6. The comparison over the different lakes showed that the GEDI beam 6 exhibits the lower bias std (below 0.10 m), much lower than the ones from beams 3 and 7 (around 0.15 m), ranking second and third in terms of low std bias. The other beams exhibit bias std between 0.15 and 0.3 m. An issue on range determination, and as a consequence the absolute elevations, affects beams 0 and 1 accounting for their low reliability for water level estimates. It has already been documented [79]. Over Lakes Geneva and Neuchâtel, a sufficient number of valid GEDI observations was available to also compute RSME and R (Figure 8). Quite large RMSE

values, ranging from 0.16 to 0.33 m over Lake Geneva, and between 0.29 and 0.51 m over Lake Neuchâtel, can be observed, larger than the ones obtained with the recent RA missions (SARAL, JASON-3, and SENTINEL-3A and B). Even if good correlations, higher than 0.75, were observed for some beams (0, 1, 6, and 7), very low correlations were obtained for the remaining beams over Lake Leman, and no correlation was found over Neuchâtel Lake. Different reasons can account for this discrepancy:

- RMSE and R were computed over a larger number of observations using RA data than using GEDI ones. In addition, as Lakes Geneva and Neuchâtel are characterized by a low seasonal amplitude (below 1 m), this could explain the lower statistical results obtained using the GEDI data, especially as GEDI sampling period is limited to 6 months.
- The footprint of the GEDI mission is much smaller (25 m of diameter) than the ones from the RA missions (a few kilometers of radius decreasing as the frequency increases from Ku to Ka bands for LRM missions, a surface of a few kilometers of length by a couple of hundreds of m of width SAR missions). GEDI waveform are more impacted by both geophysical and anthropogenic factors affecting the lake surface.
- Contrary to RA measurements, lidar observations are strongly impacted by the presence of water in the atmosphere, degrading the accuracy of the height retrievals.

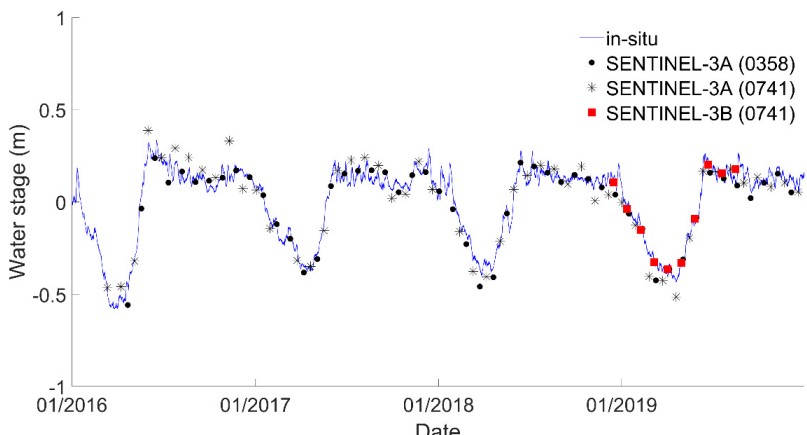

**Figure 10.** Time series of lake water level anomalies derived from SENTINEL-3A and B over Lake Geneva.

To better understand the reasons of these discrepancies, bias and RMSE were computed for every GEDI pass over eight Swiss Lakes (Zürichsee and Obersee are considered together to increase the number of overpasses and Lake Sarnen is excluded as there are too few measurements available) in this study. Not surprisingly, lower bias and RMSE values were obtained for water levels derived from beams 6 and 7, confirming that they are the more accurate ones for land hydrology (Figure 11). Considering the temporal changes, it can be seen that large deviations of the bias and high RMSE values are observed in spring and fall, which can be attributed to the presence of water in the atmosphere as they correspond to rainy periods over western Europe. However, over a few lakes (e.g., Lakes Leman and Lucerne), large RMSE values are also present during summertime. They can be related to geophysical effects including waves, small currents generated by thermal effects [80] or winds [81], seiches, and anthropogenic activities as the presence of boats on the lake surface has more impact on GEDI waveforms than on RA ones. Based on these results, GEDI data from beam 6 exhibits a strong potential for the levelling of in situ stations over lakes and river basins with unlevelled in situ stations, estimating along-track profiles of a water body which can be used either for deriving river slopes and for identifying geoid undulations, and for complementing the current RA-based water levels once inter-mission biases will be well determined.



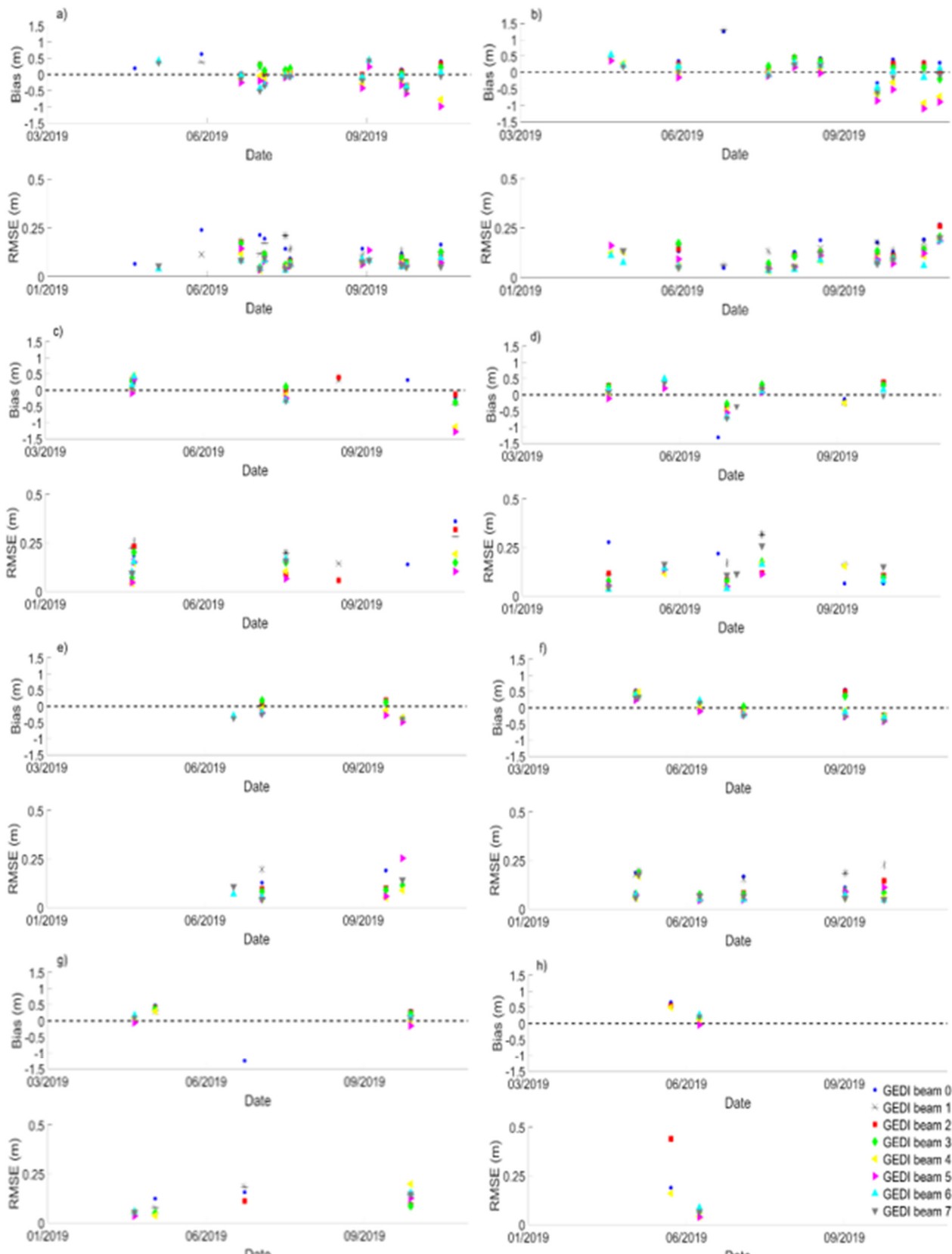

**Figure 11.** Bias (m) and RMSE (m) between in situ water levels and GEDI-based ones for every overpass and every year over (**a**) Lake Geneva, (**b**) Lake Neuchâtel, (**c**) Lake Thun, (**d**) Lake Lucerne, (**e**) Zürichsee including Obersee, (**f**) Lake Zug, (**g**) Walensee, (**h**) Lake Sempach.

## 5. Conclusions

In this study, the performances of both RA and lidar missions were assessed over monitored lakes located in a mountainous environment. To do so, the recently developed AlTiS software was used as it is able to process the data from the different RA missions presented in the manuscript. It will be maintained and regularly updated to improve it, integrate GDR from other missions, and take into account suggestions from the users. For instance, it would be possible to read and process the ENVISAT (v3), Cryosat-2 (baselines C and D), and SENTINEL-6/Jason-CS GDR in the future versions of AlTiS. The presence of hills and mountains in the surrounding of lakes has a strong impact on the capability of RA to detect the lakes surfaces. Classical LRM RA missions operating in open loop mode at Ku-band (i.e., ERS-2, ENVISAT, and Jason-1) were strongly affected by this environment, the radar remaining locked on the top of the hills and mountains causing a lack of acquisition over the lake surface, even when the altimeter ground-track was crossing the lake for several kilometers. A continuous monitoring of the lake levels was possible for less than 45%/30% of the ERS-2/ENVISAT ground-tracks. On the same orbit, SARAL was able to continuously monitor the same number of lakes as ERS-2 but outperforms both ERS-2 and ENVISAT in terms of bias, RMSE and R when compared to in situ gauge records owing its larger bandwidth and its smaller footprint. Considering the 10-day orbit, on the only crossing between Jason-1/2/3 ground-tracks and the Lake Geneva, no water stage retrieval was possible using JASON-1 data and lower RSME (0.13 against 0.21 m) and higher R (0.75 and 0.61) were obtained for JASON-3 retrievals against JASON-2 ones because Jason-3 is operating in close-loop (DEM tracking) mode. Results obtained with SENTINEL-3-based water stages outperforms the results obtained with the other missions (RMSE below 0.1 m and R above 0.85 except over Lake Lucerne) even for crossings of small width (1.5 km over Obersee and 1.1 over Walensee). In addition, SENTINEL-3A and B exhibits a very similar bias over the different lakes offering the opportunity to combine the water levels from the two satellites without taking into account a bias between their estimates. These results show the strong interest of the new operating modes (close-loop and SAR) to accurately retrieve water levels in complex environments. When available, owing to their high accuracy (RMSE $\leq 0.06$ m and R $\geq 0.95$) and constant bias (0.42 $\pm$ 0.03) m, ICESat-2 data demonstrated a strong potential for water level monitoring over land surfaces. The GEDI data need to be analyzed over longer time periods to clearly determine their capability for land surface hydrology.

**Author Contributions:** Conceptualization, F.F. and N.B.; methodology, F.F., J.-F.C. and N.B.; software, F.B., I.F. and M.B.-N.; validation, F.F., I.F. and M.B.-N.; writing—original draft preparation, F.F., N.B. and I.F.; writing—review and editing, all co-authors; funding acquisition, F.F. and J.-F.C. All authors have read and agreed to the published version of the manuscript.

**Funding:** This research was funded by CNES TOSCA grant Hydroweb and CNES/EUMETSAT TOSCA grants CTOH and FOAM" and "The APC was funded by CNES TOSCA grant CTOH".

**Institutional Review Board Statement:** Not applicable.

**Informed Consent Statement:** Not applicable.

**Data Availability Statement:** Radar altimetry data can be found here: http://ctoh.legos.obs-mip.fr/applications/land_surfaces/altimetric_data/altis/altis (accessed on 3 June 2021). GEDI data can be found here: https://lpdaac.usgs.gov/products/gedi02_av001/ (accessed on 3 June 2021). ICESat-2 data can be found here: https://nsidc.org/data/atl13/versions/3 (accessed on 3 June 2021). In-stu water levels can be found here: https://www.hydrodaten.admin.ch/ (accessed on 3 June 2021). The Swiss goid model can foun found here: https://www.swisstopo.admin.ch/en/knowledge-facts/surveying-geodesy/geoid.html (accessed on 3 June 2021).

**Acknowledgments:** We thank Michèle Oberhänsli from BAFU for her help with the description of the water stage of the Swiss lakes. We also thank two anonymous Reviewers for their comments which helped us to improve the quality of the manuscript. AlTiS is an Open Source project under CeCill license (IDDN certification: IDDN.FR.010.0121234.000.R.X.2020.041.30000—https://www.iddn.org/cgi-iddn/certificat.cgi?IDDN.FR.010.0121234.000.R.X.2020.041.30000, accessed on 3 June 2021).

**Conflicts of Interest:** The authors declare no conflict of interest. The funders had no role in the design of the study; in the collection, analyses, or interpretation of data; in the writing of the manuscript, or in the decision to publish the results.

## Appendix A

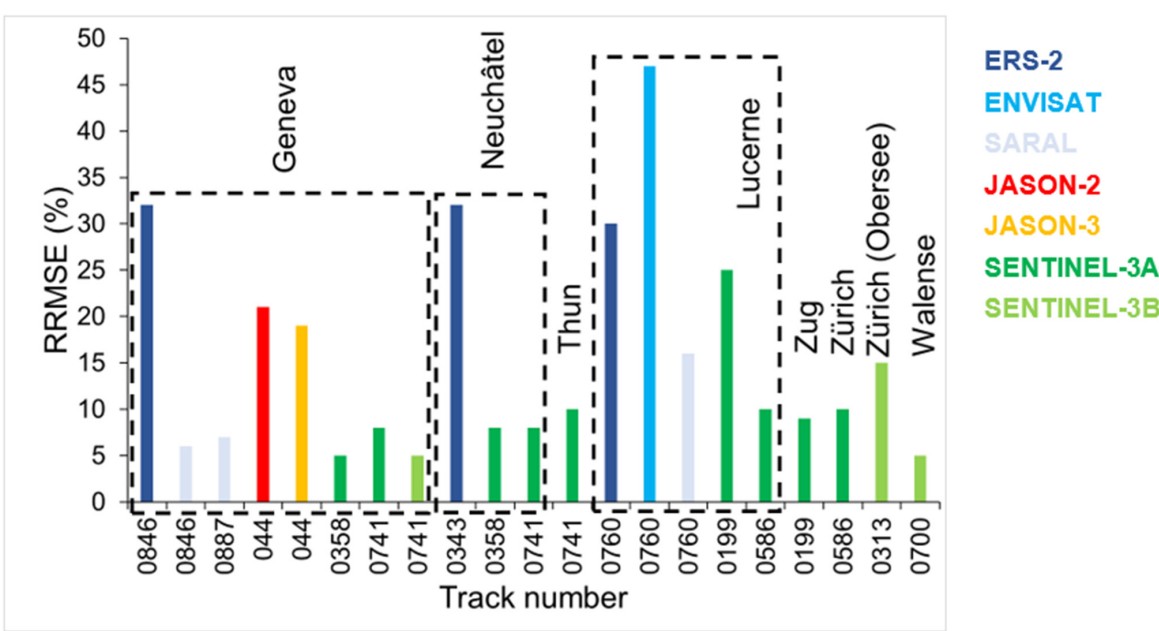

**Figure A1.** RRMSE (%) between the water levels measured in-situ and derived from RA observations over 8 Swiss lakes.

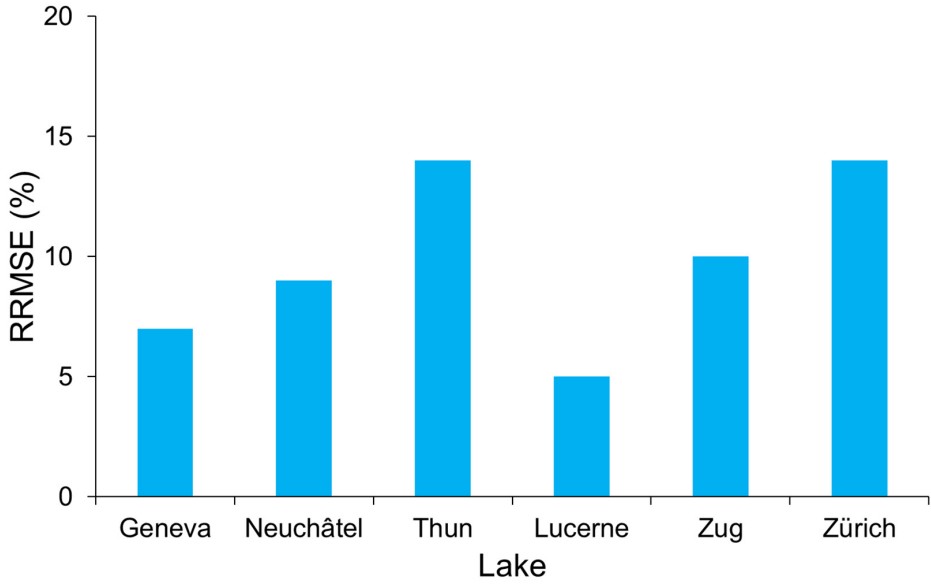

**Figure A2.** RRMSE (%) between the water levels measured in-situ and derived from ICESat-2 observations over 6 Swiss lakes.

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
