# Peer review of "Evaluation of the Performances of Radar and Lidar Altimetry Missions for Water Level Retrievals in Mountainous Environment: The Case of the Swiss Lakes"

_remotesensing, doi:10.3390/rs13112196_

Round 1
Reviewer 1 Report
The paper deals with the Evaluation of the performances of radar and lidar altimetry missions for water level retrievals in the mountainous environment. The authors processed and analyzed a significant volume of data. In my opinion, this study can be published. It is not clear what the originality of the study is. The obtained results are generally interesting. Please see some comments below: In the abstract is missing about the clear goal. What's new the present study is contributing? Please, clarify this aspect. Thus, my recommendation is a minor revision.
Author Response
We thank Reviewer1 for his/her godd appreciation of our work.
There are 3 new aspects which are new in our study. The first evaluation of ICESat-2 and GEDI data for hydrological monitoring, to our knowledge. A complete evaluation of most of all existing radar altimetry datasets in a mountainous area to clearly evaluate the interest of SAR and DEM acquisition modes. The last point is the presentation of AlTiS.
We modified the abstract as follows :
« The novelty of this study is to provide a comprehensive evaluation of the performances of the past and current radar altimetry missions according to their acquisition (Low Resolution Mode or Synthetic Aperture Radar) and tracking (close or open loop) modes, and acquisition frequency (Ku or Ka) in a mountainous area where tracking losses of the signal are likely to occur, as well as of the recently launched ICESat-2 and GEDI missions. this studyTo do so, we evaluate the quality of water level retrievals from most radar altimetry missions launched after 1995 over 8 lakes in Switzerland, using the recently developed ALtimetry Time Series software, to compare the performances of the new tracking and acquisition modes and also the impact of the frequency used », lines 27-34.
We also added these sentences to the conclusion :
« It will be maintained and regularly updated to improve it, integrate GDR from other missions, and take into account suggestions from the users. For instance, it would be possible to read and process the ENVISAT (v3), Cryosat-2 (baselines C and D) and SENTINEL-6/Jason-CS GDR in the future versions of AlTiS», lines
Reviewer 2 Report
The authors present a comprehensive study of the water level measurements by altimetry methods in Swiss lakes. The methodological findings and critical analysis of performance of different missions are of general value. The paper is well-written and illustrated. The referee supports its rapid publication as is.
List of very few errors is given below
- Fig.9b. Please, correct the colormap figures;
- Ref. 10 in the List of references. Names of the authors are duplicated;
- L.736. Please, capitalize "Issykkul";
- L.856. Initials are likely duplicated
Author Response
We thank Reviewer2 for his/her godd appreciation of our work.
List of very few errors is given below
- Fig.9b. Please, correct the colormap figures;
The colormap is right. In AlTiS, we choose to keep the altimetry data from the GDR in their original unit. For ENVISAT v2.1, the units for the altitude of the orbit, the range and is corrections are mm. Maybe, we will correct this in the future version for homogeneity purpose. To make this clear, we added the following sentence to the legend :
« The altimeter height are provided in (m) in (a) and (c) an in (mm) in (b) respecting the units provided in the GDR files. Note that in the new version of the ENVISAT GDR (v3), they are provided in m contray to the current v2.1. It will be possible to read and process this new dataset in the new version of AlTiS (1.9) available soon ».
- Ref. 10 in the List of references. Names of the authors are duplicated;
Thank you. Corrected. I did not pay attention to the dupplication when adding this reference in the reference manager.
- L.736. Please, capitalize "Issykkul";
Done
- L.856. Initials are likely duplicated
Thank you. Corrected. I did not pay attention to the dupplication when adding this reference in the reference manager.